# phastSim: Efficient simulation of sequence evolution for pandemic-scale datasets

**Nicola De Maio**[1☯*], **William Boulton**[1☯¤a], **Lukas Weilguny**[1], **Conor R. Walker**[1,2¤b], **Yatish Turakhia**[3], **Russell Corbett-Detig**[4,5], **Nick Goldman**[1]

**1** European Molecular Biology Laboratory, European Bioinformatics Institute, Hinxton, United Kingdom, **2** Department of Genetics, University of Cambridge, Cambridge, United Kingdom, **3** Department of Electrical and Computer Engineering, University of California San Diego, San Diego, California, United States of America, **4** Department of Biomolecular Engineering, University of California Santa Cruz, Santa Cruz, California, United States of America, **5** Genomics Institute, University of California Santa Cruz, Santa Cruz, California, United States of America

☯ These authors contributed equally to this work.
¤a Current address: School of Computing Sciences, University of East Anglia, Norwich, United Kingdom
¤b Current address: New York Genome Center, New York, New York, United States of America
* demaio@ebi.ac.uk

**Data Availability Statement:** The code and data used for this project are available at https://github.com/NicolaDM/phastSim. phastSim can be easily installed across most platforms (see PyPI

## Abstract

Sequence simulators are fundamental tools in bioinformatics, as they allow us to test data processing and inference tools, and are an essential component of some inference methods. The ongoing surge in available sequence data is however testing the limits of our bioinformatics software. One example is the large number of SARS-CoV-2 genomes available, which are beyond the processing power of many methods, and simulating such large datasets is also proving difficult. Here, we present a new algorithm and software for efficiently simulating sequence evolution along extremely large trees (e.g. > 100, 000 tips) when the branches of the tree are short, as is typical in genomic epidemiology. Our algorithm is based on the Gillespie approach, and it implements an efficient multi-layered search tree structure that provides high computational efficiency by taking advantage of the fact that only a small proportion of the genome is likely to mutate at each branch of the considered phylogeny. Our open source software allows easy integration with other Python packages as well as a variety of evolutionary models, including indel models and new hypermutability models that we developed to more realistically represent SARS-CoV-2 genome evolution.

## Author summary

One of the most influential responses to the SARS-CoV-2 pandemic has been the widespread adoption of genome sequencing to keep track of viral spread and evolution. This has resulted in vast availability of genomic sequence data, that, while extremely useful and promising, is also increasingly hard to store and process efficiently. An important task in the processing of this genetic data is simulation, that is, recreating potential histories of past and future virus evolution, to benchmark data analysis methods and make statistical inference. Here, we address the problem of efficiently simulating large numbers of closely

repository https://pypi.org/project/phastSim/) using the pip installer.

**Funding:** NDM, WB, LW, CRW, and NG were supported by the European Molecular Biology Laboratory (EMBL), https://www.embl.org/. CRW was funded by the National Institute of Health Research (NIHR https://cambridgebrc.nihr.ac.uk) Cambridge Biomedical Research Centre, grant number IS-BRC-1215- 20014. R.C.-D. was supported by funding from the Schmidt Futures Foundation https://schmidtfutures.com/, by an Alfred P. Sloan foundation https://sloan.org/ fellowship, and by NIH/NIGMS https://www.nigms.nih.gov/ grant R35GM128932. The funders had no role in study design, data collection and analysis, decision to publish, or preparation of the manuscript.

**Competing interests:** The authors have declared that no competing interests exist.

related genomes, similar to those sequenced during SARS-CoV-2 pandemic, or indeed to most scenarios in genomic epidemiology. We develop a new algorithm to perform this task, that provides not only computational efficiency, but also extreme flexibility in terms of possible evolutionary models, allowing variation in mutation rates, non-stationary evolution, and indels; all phenomena that play an important role in SARS-CoV-2 evolution, as well as many other real-life epidemiological scenarios.

This is a *PLOS Computational Biology* Methods paper.

## Introduction

Sequence evolution simulation is important for many aspects of bioinformatics [1]. Its most ubiquitous applications are for testing and comparing the performance of various essential tools (such as alignment, phylogenetic, and molecular evolution inference software, see e.g. [2–4]). However, simulating sequence evolution is also often used for testing hypotheses (e.g. [5]) and for inference, either for example through Approximate Bayesian Computation [6, 7], see [8, 9], or, more recently, using deep learning, see [10–12].

Many simulators address the task of simulating gene trees, or ancestral recombination graphs, as well as simulating evolution along these trees (e.g. [13–16]). Instead, here we focus on the problem of generating sequences given an input tree, as done by "phylogenetic" simulators (e.g. [17–19]). Realistic simulation of sequence evolution along a phylogenetic tree is essential, for example, for assessing and improving our methods for inference of SARS-CoV-2 phylogenies, which is a largely still open problem [20]. One important factor is the large numbers of available genome sequences for SARS-CoV-2 ($> 3,000,000$ in the GISAID database [21] as of September 2021). Despite this, there are currently no available simulation frameworks capable of simulating the scale and complex evolutionary features of SARS-CoV-2 and similar genome datasets. For this reason, we focus on the issue of simulating realistic substitution patterns for large datasets of closely related samples, as broadly observed in genomic epidemiology sequence data, and for arbitrarily complex substitution and indel models.

Here we show that sequence simulation for such large numbers of genomes is exceedingly computationally demanding for existing software. Complex evolutionary models, for example codon substitution models and rate variation, can cause significant further slow-downs. Furthermore, many existing methods do not allow the simulation of mutational patterns realistic for SARS-CoV-2, such as non-stationary and highly variable mutational processes [22–24], or don't allow the simulation of indels. We propose a new approach to efficiently simulate the evolution of many closely related genomes along a phylogenetic tree and under general sequence evolution models. Our approach simulates one mutation (substitution or indel) at a time using the Gillespie method [25], and is further tailored to reduce time and memory demand by efficiently representing and storing information regarding non-mutated positions of the genome. Furthermore, we use a multi-layered search tree structure to efficiently sample mutation events along the genome even when each position has its own mutation rate, and to efficiently traverse the phylogenetic tree and avoid redundant operations. Our approach empowers extremely flexible and fine-grained evolutionary models. For example, non-stationary models are specifiable, with each nucleotide position of the genome assigned a distinct

mutational profile, and each codon a distinct nonsynonymous/synonymous rate ratio. Similarly flexible indel models are also specifiable.

## Materials and methods

We consider the problem of simulating evolution of a DNA (or RNA) sequence along a specified input phylogenetic tree, and under a given evolutionary model. Our simulation approach is based on the Gillespie method [25], as is typically used in molecular evolution simulators [18, 19]. We assume that each position of the genome (either nucleotide or codon) evolves independently of the others, and under a time-homogeneous substitution process; that is, the rates of evolution at each position are initially specified by the user or are sampled randomly by the simulator. We focus on the efficient simulation of sequence evolution for large phylogenetic trees with short branches: we assume that only a few mutations happen on each branch across the genome, which is typical for genomic epidemiology, and in particular for SARS-CoV-2 [26].

### Simple approach

If we assume that evolutionary rates are homogeneous across the genome, it is simple to use the Gillespie approach efficiently in this scenario by adopting an efficient representation of ancestral genomes in terms of differences with respect to a root genome [27]. As a very simplified example, let's consider the case in which there is no selective force at play, mutation rates are constant across the genome, there are no indels, and all bases mutate into all other bases at the same rate (JC69 model [28] with equal nucleotide frequencies). Throughout the manuscript, we will not assume equilibrium or stationarity in sequence evolution, but instead assume that we are given a genome at the root of the phylogeny, which we then evolve down the tree according to given rates.

In this simplified scenario, the total mutation rate across the genome is equal to the mutation rate for one base, $3r$, times the genome length (which we assume constant), $L$. Starting from the root and its genome, we visit each branch of the tree one at the time in preorder traversal. For each branch of the tree, we consider its length $t_b$, and we recursively sample a time for the next mutation from an exponential distribution with rate parameter $3rL$. If the sampled time $t$ is over $t_b$, we move to the next branch. Otherwise, we decrease $t_b$ by $t$ and we sample a mutation event. In the considered scenario, this simply means sampling one position of the genome at random (a random integer number $1 \leq i \leq L$), and then a random allele $b$, different from the current allele at position $i$, to mutate into. Additional steps are also required to keep track of mutations which have already occurred and allow them to further mutate, for example, possibly reversing a mutated allele back to the reference allele. We track each sampled mutation by adding it to a list of mutations for the current branch. It is worth noting however that there are more efficient ways to keep track of mutations that have already occurred, which we discuss in subsequent sections.

A pseudocode description of the algorithm is given in Algorithm 1. So overall, the total cost of this simple algorithm is constant in genome size, and is linear in the number of tips $N$. It does however scale with the number of mutation events (total tree length) $M = O(Nl)$ where $l$ is the average number of mutations per branch. The initialization step has cost $O(N)$ in order to read the phylogenetic tree, and further $O(L)$ with more complex models in order to keep track of the positions of different alleles. Performing the simulations has cost $O(M \log(N) + M^2 \log(N)/N) = O(l^2 N \log(N))$; the main cost here is to screen previous mutation events at each new mutation, and this can be significantly reduced as explained in the next section. There is a caveat however. The default output of our software phastSim is a concise representation of the

simulated sequences. If, however, we want to produce a file containing the full alignment in FASTA or PHYLIP formats, the memory and time cost of the algorithm will become $O(NL)$.

**Algorithm 1** Simple algorithm for one phylogenetic branch.

```
Here evolution on one branch is considered. t_b is initialized as the
length of the considered branch. r is the mutation rate from one
nucleotide to any other nucleotide. L is genome size. ref[i] is the
reference allele at position i. "Node" is the child node of the cur-
rently considered branch.
  Sample t (the time to next mutation event) from an exponential dis-
tribution with rate parameter 3rL.
  while t < t_b do
    Sample a random integer 0 < i ≤ L
    if i is not a position previously mutated in an ancestor of Node
then
      a ← ref[i].
    else
      a is the current allele for Node
    end if
    Sample a random new allele b ≠ a.
    Add mutation (i, a, b) to the list of mutations of Node.
    Update current allele for Node at position i as b.
    t_b ← t_b − t
    Sample t from an exponential distribution with parameter 3rL.
  end while
```

In classical implementations of sequence evolution simulators [17], for each node of the tree we need to update each base of the genome one at the time, therefore incurring in cost $O(NL)$. Therefore, when the number of expected mutations is $M \ll NL$ we expect an advantage in using this approach.

A considerable limitation of the above simple approach is that we assume that rates are the same across the genome, and this is hardly realistic [29, 30]. We implemented an extension of this algorithm above which accounts for both an arbitrary nucleotide substitution model (UNREST [31]), and for rate variation across the genome in terms of a finite number of rate categories. To achieve this, we extended the algorithm above to keep track of which positions of the genome have which rates. This allows us to efficiently calculate total mutation rates for each class of sites, and to efficiently sample sites within a class.

We also implemented a new model of rate variation in order to better fit the patterns of hypermutability observed in SARS-CoV-2. In this model, small proportions of hypermutable sites are given a (possibly much) higher mutation rate. At an hypermutable site, only one specific mutation rate (from one nucleotide to one other nucleotide) is enhanced. For example, one such site with hypermutability might have only the G→T mutation rate increased 100-fold, while all other rates at that site remain the same. This is to model the effects observed in SARS-CoV-2 which are possibly attributable to APOBEC and ROS activity (or other still unclear mechanisms) [22, 23].

However, as the number of site classes increases, and as the number of alleles increase (for example when considering codon models), the efficiency of the extension of the simple approach described above deteriorates, especially when each site of the genome is given different evolutionary rates. For this reason, we developed a more complex ("hierarchical") algorithm that remains efficient in light of rate variation, with only a small efficiency sacrifice relative to the simple method in the scenario of no rate variation. We allow phastSim users to choose between the simple approach or the more complex hierarchical one, that we describe below. Advanced features, for example simulation of indels, are only implemented with the hierarchical algorithm.

### Hierarchical approach

**Binary genome search tree.**   We first describe the structure and algorithm that allow us to efficiently sample a mutation event along the genome when each position might have a distinct mutation rate. This structure needs to be efficiently updatable following a mutation event; in fact, a mutation event changes the allele at a position of the genome, and therefore also its mutation rate. This is very similar to the problem of sampling from a categorical distribution with many elements, where the probabilities can be slightly modified at each sample [32]. A Huffman tree [33] would be close to optimal for this task, however, here we implement a binary search tree, which has a slightly higher expected cost [32] but allows us to more efficiently model blocks of contiguous nucleotides, and therefore to efficiently simulate indels.

In our "genome" search tree (which is distinct from, and should not be confused with, a phylogenetic tree), each node corresponds to a contiguous block of nucleotides along the genome. The root node represents the whole genome, and contains a rate value corresponding to the global mutation rate of the whole genome. The two children of the root correspond to the first and the second half of the genome, respectively. There is no overlap between the regions considered by each child node, and their union gives the region considered by the parent node. Consequently, the sum of the rates of the children of a node is equal to the rate of the node. Given this structure, we also refer to this binary search tree as the "genome" tree. A terminal node of the genome search tree corresponds to one unit of the genome, either a base or a codon, depending on the model we choose for simulations. A terminal node contains not only information about the position of the unit along the genome, but also the reference allele at this position and the mutation rates associated with it, to allow sampling of a specific mutation event at the given position/node. A graphical representation of an example genome search tree is depicted in Fig 1.

Sampling a mutation time is done as in the simple approach: sampling from an exponential distribution with rate parameters determined by the total mutation rate at the root of the genome search tree. Then, to sample a specific substitution event at a specific genome position, we first sample a random value uniformly in [0, 1) and multiply it by the total mutation rate $R$. Then, we traverse the tree from the root to the terminal node corresponding to the mutated position, which takes $\log(L)$ time. Finally, once reaching the corresponding terminal node (genome position) we choose a random substitution event affecting this position and correspondingly a new allele $a$ for this position. An example mutation sampling is depicted in Fig 1. A pseudocode description of this algorithm is given in Algorithm 2. The cost of this approach is linear in the number of alleles, making it much more efficient than classical simulation methods based on matrix exponentiation when large state spaces (e.g. codon models) are considered. Furthermore, the computation cost for simulating under a codon model can be further reduced by considering that typically a codon model only allows a maximum of 9 substitution events from any codon, so at each terminal node we only need to consider a maximum of 9 events and rates at any time. Thanks to this, the cost of running a codon model with this approach is similar to the cost of running a nucleotide model.

**Algorithm 2** Sampling of a substitution event along a genome search tree, and updating the genome search tree.

```
We assume we are given node "Root", the root of a genome search tree
structure. For any given genome node, "Node", Node.rate represents the
corresponding total subtree rate. Node.children represents the list of
its child nodes (2 children for internal nodes, 0 for terminal nodes).
Node.parent is the parent node of "Node". Node.allele is its current
allele. Node.rates (which is only allocated for terminal nodes) repre-
sents the 2-dimensional matrix of mutation rates (from any allele to
```

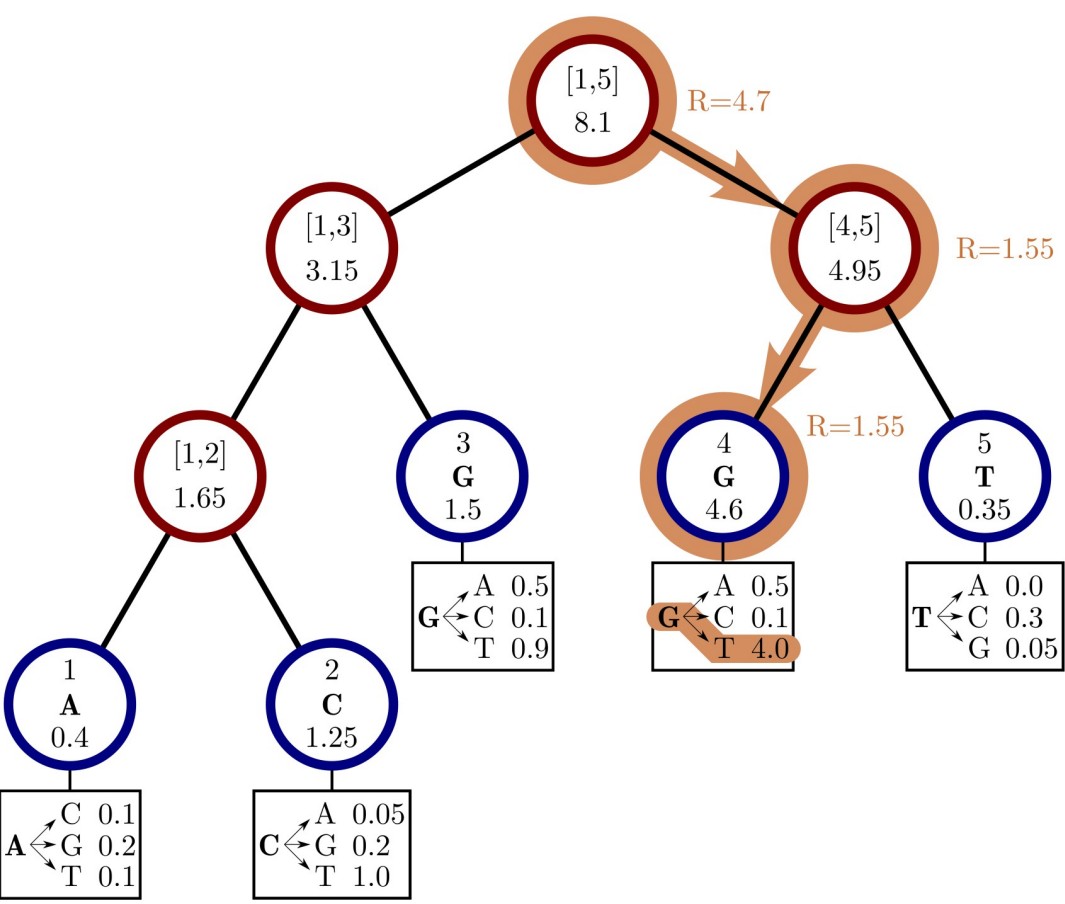

**Fig 1. Example genome search tree and its use.** An example genome search tree for ancestral genome ACGGT. Blue nodes are terminal and red nodes are internal. Inside each node we represent on top the genome positions represented by the node; at the center inside terminal nodes we show the allele of the node; at the bottom of nodes is their total rate. Under each terminal node we show the example relevant mutation rates. The path highlighted in orange shows an example sampling of one mutation event. A parameter $R$ is assigned an initial random number sampled uniformly between 0 and the total rate 8.1, in this case it is $R = 4.7$. As we move downward, the value of $R$ can decrease, as described in Algorithm 2, determining which site will mutate and how. Here, an initial $R = 4.7$ results in the sampling of a G→T mutation at genome position 4.

any other allele) at the given position, and for simplicity here we assume that a rate of an allele into itself (a matrix diagonal entry) is 0; for codon models, for efficiency the rows of the matrix are only allocated and filled when they are needed for the first time. Node. position (only defined for terminal nodes) refers to the genome position represented by the node. The list "mutations" is used to record the mutation events simulated on the considered phylogenetic branch. The function "sample(rates)" samples a rate from a list proportional to its value. We are also given a random number $0 \leq R <$ Root.rate for which we want to sample the corresponding mutation event.

```
  Node ← Root
 while Node is not terminal do
   for Child in Node.children do
     if Child.rate < R then
       R ← R – Child.rate
     else
       Node.rate ← Node.rate – Child.rate
```

```
        Node ← Child
        break
      end if
    end for
  end while
  oldAllele ← Node.allele
  Node.allele ← sample(Node.rates[oldAllele])
  mutations.append([Node.position,oldAllele,Node.allele])
  Node.rate ← ∑_b Node.rates[Node.allele][b]
  NewRate ← Node.rate
  while Node.parent is not null do
    Node ← Node.parent
    Node.rate ← Node.rate + NewRate
    NewRate ← Node.rate
  end while
```

As mentioned before, once a mutation event is sampled, we need to modify the sampling process so that the change in allele at the mutated position is taken into account, since this change usually affects local and global mutation rates (a rare exception is for example when substitution rates are all equal). Modifying our genome search tree following a substitution event is both simple and efficient: we simply need to modify the rates and allele at the mutated terminal node, and then update the rate of all ancestors of this terminal node accordingly. Algorithm 2 for example describes how to sample a substitution event from a genome search tree as well as how to update the genome search tree accordingly. Again, this can be done in $\log(L)$ time for each new substitution event sampled. However, while this is efficient for simulating evolution along a temporal line, that is, along a single branch of the phylogeny descendant from the root, it becomes inefficient for simulating evolution along a phylogenetic tree. This is because, if we modify the tree, then we cannot use it as it is for the sibling nodes. In other words, when we reach a split in the phylogenetic tree, and we have two children of the same phylogenetic node, we need to pass the same genome search tree of the phylogenetic parent node to both phylogenetic children. However, we can't only pass a pointer to the same tree to both children, because evolving along one branch leading to one sibling would modify the genome search tree also for the other sibling. If we take the approach of duplicating the genome search tree at each phylogenetic split, we end up with a cost $O(NL)$, which we are trying to avoid. For this reason, we devise an alternative, hierarchical, multi-layered approach to evolving a genome search tree, described below. Later on in the text we also describe the extension of our approach and of the genome search tree structure to simulate indels.

**Hierarchical, multi-layer evolving genome search tree.**   In order to use our genome search tree structure to sample mutations along a phylogenetic tree, we add a further "vertical" dimension to it. At each branch of the phylogenetic tree, instead of modifying a genome search tree, we take the approach of building on it, without modifying the starting genome search tree nodes, so that the original genome search tree is not lost but instead is preserved at "layer 0" of our multi-layer structure. When we sample a mutation, we create a few new genome search tree nodes in the corresponding layer of the structure, instead of not an entire new genome search tree. By doing this, we can effectively adapt a (multi-layer) genome search tree as new mutations are sampled without losing the original genome search tree. This means that when we can pass the same genome search tree to two children of a phylogenetic node without needing to duplicate the genome search tree structure. Instead, we simply remove (de-allocate, or ignore) the genome search tree nodes that have been added to other layers by the descendants of the first child node, and pass the same genome search tree structure to the two considered child nodes. A graphical representation of an example multi-layer genome search tree and its evolution is given in Fig 2.

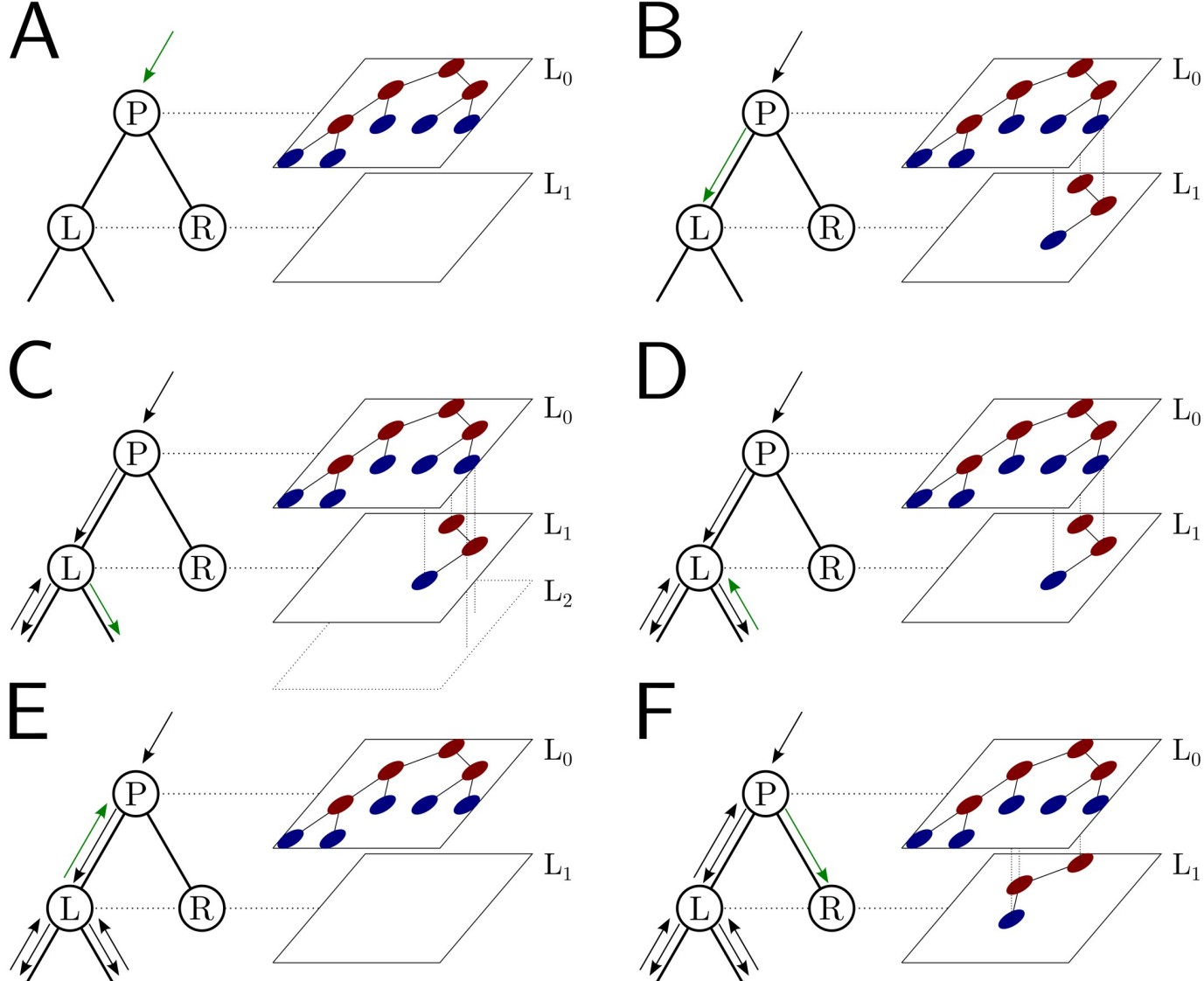

**Fig 2. Example of multi-layer genome search tree and its evolution.** We track the evolution of the multi-layer genome search tree starting from the genome search tree of Fig 1. Colors for the genome search tree are the same as in Fig 1 (right side of each panel). On the left side of each panel, we show an extract of the phylogenetic tree containing three nodes ("P" for parent, which in this example is the root of the phylogeny, and "L" and "R" for left and right node). "L" has further descendants, but we don't show them here and only focus on this triplet of nodes as an example. The green arrow along the phylogenetic tree shows the current step of the preorder traversal being considered by the given panel. Black arrows show past steps. Vertical dashed lines in the multi-layer genome search tree connect nodes that represent the same portions of the genome but that are in different layers. "L0" stands for "Layer 0" and "L1" for "Layer 1", etc. **A** At the phylogenetic root "P" we initialize the genome search tree for layer 0. **B** As we move to child "L", a new substitution is sampled (as in Fig 1) and 3 corresponding genome nodes are created in layer 1. These nodes correspond to the nodes in the original genome search tree whose rate is affected by the new mutation. **C** As we traverse the subtree of the descendants of L, new nodes and mutations might be added in the layers below. **D** We are finished traversing the subtree of the descendants of L, and we return to L, at which point all nodes in layer below 1 have either been removed or have become irrelevant. **E** We return to P, at which point the genome search tree nodes previously added layer 1 are also ignored or deleted. **F** We move from P to R, and in doing so new mutation events might be sampled and the corresponding genome nodes might be added to layer 1 (new genome search tree nodes corresponding to 1 new substitution are shown in the new layer 1).

We start with a genome search tree at the phylogenetic root node; additional nodes are then added at further layers. A genome search tree layer $n$ represents the genome nodes specific to a particular depth of the phylogenetic tree; phylogenetic nodes closer to the root (in terms of number of branches that need to be traversed from the root) will be associated with a lower $n$,

and those more distant from the root with higher $n$. All the initial nodes of the original genome search tree belong to layer 0, the layer corresponding to the phylogenetic root. Then, as we move from the phylogenetic root to its first child, we add nodes to the tree in layer 1, representing the consequences of mutation events happening along the branch between the phylogenetic root and the first child. Nodes in layer 0 only point to nodes in layer 0, and never to nodes in other layers. More generally, nodes in layer $n$ only point to nodes in layers $m \leq n$. Every time the multi-layer genome search tree is passed from phylogenetic parent (layer $n$) to child (layer $n + 1$), new nodes are added to the corresponding layer ($n + 1$) if mutation events occur on the corresponding phylogenetic branch.

We traverse the phylogenetic tree in preorder traversal, so, starting from the root, we move to the first child, to which we pass the initial genome search tree, add new layers, then do the same for this child's children. For each new mutation occurring on this phylogenetic branch connecting the root and its first child, we traverse the genome search tree, and every time we would modify the genome search tree (to update the mutation rates following a change of allele at a position) we instead create new genome search tree nodes in the child layer. Once we have traversed the whole phylogenetic subtree of the first child of the root, we have to move to second child of the root. This operation does not incur the cost of duplicating any part of the genome search tree, as we only need to pass to the second child the pointer to the root of layer 0 of the hierarchical genome search tree. Similarly, at any internal phylogenetic node at layer $n$, to both children we pass the pointer to the root of layer $n$ of the genome search tree. The only additional step which might be required is the de-allocation of nodes in layer $n + 1$ as we move from one node to its sibling (thanks to our preorder traversal, the nodes currently in and below this layer will not be used again), but this step at most only slows simulations by a small constant factor.

At the start of the simulations for each branch, moving from layer $n$ to $n + 1$, we first create a new genome root node for layer $n + 1$. This root initially points to the same children as the genome root at layer $n$, and it also has the same total rate. After creating a new layer root, we sample mutation events for the current phylogenetic branch. To sample mutations, we follow the binary search tree determined by the root of layer $n + 1$. As a new mutation event is picked, we either create new layer $n + 1$ nodes, or modify existing layer $n + 1$ nodes. When sampling a new mutation, every time we reach a node in the genome search tree, we either modify the rate of the node, if it's in layer $n + 1$, or we create a new layer $n + 1$ node, if the original node was in a different layer. The new node is given at first the same children as the original node. When a terminal node is reached, we calculate its new rates (unless they have already been created before for some other node in the phylogenetic tree, in which case we just retrieve them from a dictionary) and total rate, and we pass the new total rate to its parent node, which uses it to update its own total rate, and so on. In total, the cost of sampling a new mutation event and updating the multi-layered structure is $O(\log(L))$. A sketch of the mutation sampling process and multi-layer genome search tree update is given in Algorithm 3. The total cost of the algorithm is then $O(L + N + M \log(L))$, where the addendum $L$ is due to the initial creation of the layer 0 genome search tree, and $N$ is due to the tree traversing process. In a scenario like SARS-CoV-2 genomic epidemiology, this can lead to dramatic reduction in computational demand compared to the standard $O(NL)$ in the field, since $M$ appears typically not distant from $N$ [22–24]. We give a summary of the global hierarchical method in Algorithm 4.

**Algorithm 3** SampleMutation(Node,Layer,$R$): Sampling of a mutation event along a multi-layer genome search tree.

```
This function is initially run on the root node "Root" of a genome
search tree for layer "Layer". Parameters are as in Algorithm 2; in
addition, Node.layer represents the layer of the considered node.
```

```
While below we simplify a few details, in reality we don't recalculate
rates at every mutation, but we only calculate them the first time
they are needed, and then store them in dictionaries.
```
 **if** Node.layer ≠ Layer **then**
```
    create a new node NewNode copy of Node
    NewNode.layer ← Layer
    Node ← NewNode
```
 **end if**
 **if** Node is terminal **then**
```
    sample mutation event from Node.allele using Node.rates and R.
    expand if needed Node.rates, and update Node.rate and Node.allele
```
 **return** Node
 **else**
 **for** $c$ in Length(Node.children) **do**
```
      Child ← Node.children[c]
```
 **if** Child.rate < R **then**
```
        R ← R – Child.rate
```
 **else**
```
        Node.rate ← Node.rate – Child.rate
        NewChild ← SampleMutation(Child,Layer,R) {note that SampleMu-
tation is the current function, so this function is called
recursively}
        Node.children[c] ← NewChild
        Node.rate ← Node.rate + NewChild.rate
        return Node
```
 **end if**
 **end for**
 **end if**

**Algorithm 4** SimulatePhyloNode(Node,GenomeNode,Layer): Hierarchical algorithm for simulating sequence evolution along a branch of the phylogenetic tree.

```
Here evolution on a branch of the phylogenetic tree is considered. The
branch is passed through Node, which represents the child node of the
branch. The branch length is Node.length. Simulation of the whole phy-
logeny is performed by calling SimulatePhyloNode(Root,GenomeRoot,0),
where Root is the root of the phylogenetic tree (we assume Root.
length = 0) and GenomeRoot is the root of the initial genome search
tree for layer 0. This layer 0 genome search tree is created by consid-
ering the genome of the phylogenetic root, which is typically either
sampled at random or read from a reference genome.
  Sample t (the time to next mutation event) from an exponential dis-
tribution with rate parameter GenomeNode.rate
  CurrentTime ← t
```
 **while** CurrentTime < Node.length **do**
```
    Sample a random uniform vaule 0 ≤ R < GenomeNode.rate
    GenomeNode ← SampleMutation(GenomeNode,Layer,R) {note that this
is the function defined by Algorithm 3}
    Sample t (the time to the next mutation event) from an exponential
distribution with rate parameter GenomeNode.rate
    CurrentTime ← CurrentTime + t
```
 **end while**
 **for** Child in Node.children **do**
```
    run SimulatePhyloNode(Node,GenomeNode,Layer+1) {note that this
function is the one defined in the current Algorithm, which is there-
fore recursive in nature}
```
 **end for**
```
  if needed, de-allocate all nodes of layer Layer from GenomeNode down
to its descendants. {Because genome nodes with layer Layer are
```

```
descendant only from nodes with layer Layer, we do not need to traverse
the whole multi-layer genome search tree, but only its layer Layer.}
```

## Indels

We further extended the multi-layer genome search tree approach to efficiently simulate insertions and deletions. Each leaf on the genome search tree is assigned a deletion rate, insertion rate, and substitution rate, denoted $R_d$, $R_i$, and $R_s$ respectively, and the total mutation rate for the leaf will be $R_d + R_i + R_s$. The substitution rate $R_s$ itself is the sum of all substitution rates from the current allele of the leaf. Insertions are modeled as occurring on the right (3′ end) of the sampled position; to model insertion at the 5′ end of the genome, a dummy terminal genome search tree node is employed representing the leftmost end of the genome, and is initialized with $R_s = R_d = 0$ but with non-zero $R_i$. Just as with the substitution rates, which can be site specific, $R_d$ and $R_i$ can be drawn from a gamma distribution, or can be constant across the genome. When a mutation event is sampled at a node, it will be sampled as a deletion, an insertion, or a substitution proportionally to $R_d$, $R_i$ and $R_s$.

Our software allows for indels with lengths drawn from a number of parametric distributions following the options allowed with INDELible, see Table 1 for an overview of the various distributions that have been implemented. Sampled indels have always length $\geq 1$.

Below we explain in more detail the algorithm used to efficiently simulate insertions and deletions using multi-layered genome search trees. In short, insertion events are simulated by adding new small subtrees to the genome search tree in the current layer. Deletion events are instead simulated by setting substitution and indel rates to 0 in deleted nodes.

**Insertion algorithm.** The algorithms for simulating insertions and deletions mostly proceed as the one simulating substitutions (Algorithm 3) in that we traverse the genome search tree to find the terminal node "Leaf" affected by the next sampled mutation. We then sample the type of the next mutation event (insertion, deletion, or substitution) proportional to the corresponding mutation rates $R_i$, $R_d$ and $R_s$ of "Leaf". The process for simulating a substitution remains the same as before. If instead a new insertion event is simulated, we sample a length $l$ for the inserted material from the corresponding prior distribution, and then add a new subtree to the genome search tree as detailed in Algorithms 5 and 6.

**Algorithm 5** insertNode(Node, $l$): this function inserts a new genome subtree at the given terminal genome search tree node "Node" at which the insertion is sampled, given the insertion length $l$. We assume that Node is part of the current layer, and that new nodes are created at the current layer.

```
  insertionRootNode ← populateGenomeTree(l) {This calls algorithm 6
to generate an insertion subtree of size l.}
  Create a new genome search tree internal node newInternalNode
  newInternalNode.parent ← Node.parent
  Replace Node with newInternalNode as child of Node.parent.
  Node.parent ← newInternalNode
```

**Table 1. Indel length distribution options.**

| Distribution | Parameters | $P(X = n)$, $n > 0$ |
|---|---|---|
| Geometric | $p$ | $(1 - p)^{n-1}p$ |
| Negative Binomial | $p, k$ | $\binom{k+n-1}{n}(1 - p)^{n-1}p^k$ |
| Zeta | $a$ | $n^{-a}/\zeta(a)$ |
| Lavalette | $a, k$ | $\left(\frac{kn}{k-n+1}\right)^{-a}$ for $n \leq k$ |
| Discrete | vector $\mathbf{v}$ | $v_n$ |

```
insertionRootNode.parent ← newInternalNode
newInternalNode.children ← [Node, insertionRootNode]
newInternalNode.rate ← Node.rate + insertionRootNode.rate
```

**Algorithm 6** populateGenomeTree(*l*): this function recursively creates a new genome subtree for a given insertion of length *l*, and returns the root node of the subtree.

```
Create a new node genome search tree node Node.
if l = 1 then
  Sample Node.allele from a prior distribution (typically the refer-
ence nucleotide or codon frequencies).
  Sample mutation rates Rᵢ, R_d, R_s for Node.
  Node.rate ←Rᵢ + R_d + R_s
else
  leftL ← ⌊l/2⌋
  rightL ← l−leftL
  leftNode ← populateGenomeTree(leftL)
  rightNode ← populateGenomeTree(rightL)
  leftNode.parent ← Node
  rightNode.parent ← Node
  Node.children ← [leftNode, rightNode]
  Node.rate ← leftNode.rate + rightNode.rate
end if
return Node
```

If the user specifies a root genome, inserted nucleotides are randomly and independently sampled from the root genome nucleotide frequencies; if the user does not specifies a root genome, but instead specifies root nucleotide frequencies for phastSim to sample a random root genome from them, then the same sampling is done for inserted sequences. The substitution model for each inserted nucleotide/codon is chosen at the time of insertion in the same way as for the root genome (in particular also accounting for rate variation).

Note that the addition of new subtrees to the genome search tree will typically make it less balanced, and a potentially less efficient search tree. In typical scenarios considered here, that is, when divergence is low and all genomes are closely related to the root, the effect of this imbalance on the overall search-efficiency of the genome search tree will be extremely minor.

**Deletion algorithm.** If the next mutation event at Leaf will instead be a deletion, again, we first sample a deletion length *l*, and then we proceed to set to 0 the total mutation rate for node Leaf and its following *l* − 1 positions of the genome in the current layer, ignoring positions that are already deleted. The main subtlety of this approach is to avoid traversing the whole genome search tree (incurring a cost of $O(L)$ operations), to delete these *l* characters. Algorithm 7 below performs this task efficiently employing at most $O(l \log(L))$ operations. This algorithm returns the number of characters deleted. If Leaf represents a position near the 3′ end of the genome, the sampled deletion length might overflow past the end of the genome, and so fewer than *l* positions might be effectively deleted.

**Algorithm 7** deleteNodes(Rand,GenomeNode,Layer,RemainingDeletions): recursive algorithm for deleting nodes of the genome search tree following a deletion event. It returns the number of positions deleted. "Rand" is the random number that has been used to sample the deletion event—here it's used to direct the search to the first deleted positions and the following ones.

```
if GenomeNode.isTerminal then
  {Skip gap characters and only delete nodes with a non-gap symbol.}
  if GenomeNode.allele ≠ "-" then
    GenomeNode.allele ← "-"
    GenomeNode.rate ← 0
    return 1
  else
```

```
        return 0
      end if
  else
    deletedPositions ← 0
    totalRate ← 0
    for Child in GenomeNode.children do
      if RemainingDeletions = 0 then
        return 0
      end if
      if Rand > child.rate then
        Rand = Rand−child.rate
      else
        if child.layer ≠ Layer then
          create newChild, copy of child in layer Layer.
        else
          newChild ← child
        end if
        newDeletions ← deleteNodes(Rand,newChild,Layer,
RemainingDeletions)
        RemainingDeletions ← RemainingDeletions−newDeletions
        deletedPositions ← deletedPositions + newDeletions
        totalRate ← totalRate + newChild.rate
      end if
    end for
    GenomeNode.rate ← totalRate
    return deletedPositions
  end if
```

## Further details of the implementation

**Substitution models.**   Thanks to our algorithm, we can allow any substitution model without incurring a dramatic increase in computational demand, and without risking numerical instability (which can sometimes be a problem with classical matrix exponentiation approaches). Users can easily specify different nucleotide substitution matrices (e.g. JC [28], HKY [34], or GTR [6]). By default, we adopt the most general nucleotide substitution model, UNREST [31], using as default rates those we estimated from SARS-CoV-2 [22].

We also implemented codon models, which, with our hierarchical approach, come at only a small additional computational demand compared to nucleotide models. To define substitution rates of codon models, we use an extension of the GY94 [35] model, and separately model the nucleotide mutation process and the amino acid selection one. Unlike GY94 (which assumes an HKY nucleotide mutation process), we allow any general nucleotide mutation process as defined by an UNREST matrix. Then, nonsynonymous mutations rates are modified by a single factor $\omega$ (see next section for variation of $\omega$ across codons). Under this model, a substitution from codon $c_1$ to codon $c_2$ therefore has rate:

$$r_{c_1 \to c_2} = \begin{cases} m_{n_1 \to n_2}, & \text{if } c_1 \text{ and } c_2 \text{ are synonymous and differ only by} \\ & \text{nucleotides } n_1 \text{ and } n_2 \text{ at a position}, \\ \omega m_{n_1 \to n_2}, & \text{if } c_1 \text{ and } c_2 \text{ are non-synonymous and differ only by} \quad (1) \\ & \text{nucleotides } n_1 \text{ and } n_2 \text{ at a position}, \\ 0, & \text{if } c_1 \text{ and } c_2 \text{ differ by more than one nucleotide}, \end{cases}$$

where $m_{n_1 \to n_2}$ is the mutation rate from nucleotide $n_1$ to nucleotide $n_2$.

**Table 2. A comparison of features of different sequence evolution simulation software packages.**

|  | phastSim | Seq-Gen [17] | INDELible [18] | pyvolve [36] |
|---|---|---|---|---|
| Indels | Yes | No | Yes | No |
| Nucleotide Models | $\leq$ UNREST | $\leq$ GTR | $\leq$ UNREST | $\leq$ GTR |
| Codon Models | Extended GY94 | No | GY94-style | GY94, MutSel and MG94-style |
| Amino Acid Models | No | Yes | Yes | Yes |
| Hypermutability | Yes | No | No | No |

We don't allow, at this stage, instantaneous multi-nucleotide mutation events, or amino acid substitution models, but we plan to address them in future extensions. A description of currently implemented models and a comparison with those in other similar simulation software is given in Table 2.

**Models of rate variation.** We consider four types of variation in rates across the genome. These types can be used in combination, or separately, as required.

The first type of variation is changes in the position-specific mutation rate across the genome. Every nucleotide position $i$ in the genome (even when using a codon model) is assigned its own mutation rate scaling factor $\gamma_i$. This means that, at position $i$, the mutation rate from any nucleotide $n_1$ to any other nucleotide $n_2$ becomes $\gamma_i m_{n_1 \to n_2}$. We allow two ways to sample values of $\gamma_i$ for each $i$. One way is to sample them from a continuous Gamma distribution with parameters $\Gamma(\alpha, \alpha)$, with $\alpha$ specified by the user; this results in each genome position having a distinct $\gamma_i$. Alternatively, we allow the definition of discrete categories, with a finite number of categories, each with its own proportion of sites and $\gamma$ rate.

The second type of variation we model is variation in $\omega$, with each codon position $i$ across the genome being given its own $\omega_i$. As with $\gamma_i$, values of $\omega_i$ can either be sampled from a continuous Gamma or a finite categorical distribution.

Lastly, to accommodate the strong variation in mutation rates observed in SARS-CoV-2 [22, 23] attributable to APOBEC, ADAR, or ROS activity, we introduce a new model of rate variation. This model allows, for a certain position, to have one specific mutation rate (from one specific nucleotide to another specific nucleotide) enhanced by a certain amount $\mu$. In this case we only allow a categorical distribution, with the first category having no enhancement ($\mu = 1$) and the other categories having $\mu > 1$. For any nucleotide position $i$ that is assigned a hypermutable category and therefore has $\mu_i > 1$, we then sample uniformly a start nucleotide $n_s$ and a destination nucleotide $n_d$. The mutation rates $m_{n_1 \to n_2}^i$ for position $i$ then become:

$$m_{n_1 \to n_2}^i = \begin{cases} \gamma_i m_{n_1 \to n_2}, & \text{if } n_1 \neq n_s \text{ or } n_2 \neq n_d, \\ \gamma_i \mu_i m_{n_1 \to n_2}, & \text{otherwise}. \end{cases} \tag{2}$$

**Rate normalization.** We assume that, for the given input phylogenetic tree, branch lengths represent expected numbers of substitutions per nucleotide (no matter if a nucleotide or a codon model is used) for the root genome. As mutations accumulate across the phylogeny, the total mutation rate of the genome might slightly change; this is particularly true because we allow substitution models that are not at equilibrium. This also means that while branch lengths near the root represent the expected numbers of substitutions per nucleotide, as one moves down the tree the expected number of nucleotide substitutions expected on a branch might not be a simple function of the branch length.

**Output formats.**   As default, our software creates an output file where it stores information about which genome position evolved under which rate. It also creates a file where each tip name is listed together with the mutations it contains that distinguish its genome from the reference genome. In scenarios similar to SARS-CoV-2 datasets (where each genome is very similar to the reference), this format requires much less space and time to generate than FASTA or PHYLIP formats (see the "Simple approach" subsection).

An optional output format that our software can create is a tree in Newick format, where each branch of the input phylogeny is annotated with a list of mutation events that occurred on that branch. This format is richer than the others, as it provides information regarding each mutation event, even those that might be over-written by other mutations at the same position; it is also more efficient than multiple sequence alignment formats in the scenario of short branch lengths considered here. We also allow a binary analogue of this annotated Newick tree, called a MAT (mutation annotated tree) [37], which is compatible with the phylogenetic software UShER [27].

Finally, we also allow the creation of unaligned FASTA output. However, note that the creation of a FASTA file costs $O(NL)$ in time and space. In the case simulations are performed without indels, we also allow the generation of a PHYLIP format alignment output.

**Python package.**   Our software phastSim is implemented as a Python package, and can be found at https://github.com/NicolaDM/phastSim or https://pypi.org/project/phastSim/. phastSim uses the ETE3 library [38] to robustly read input trees in different variants of the Newick format. We tested the correctness of our simulator with a series of tests, some of which are showcased in S1 Text.

## Comparisons with other methods

To assess the performance of our approach we compare it to other popular sequence simulation methods. INDELible [18] is a sequence evolution simulation software that is particularly useful since it allows simulation of indels and evolution of codon sequences. INDELible allows simulations under two different algorithms: method 1 ("INDELible-m1") uses matrix exponentiation to model substitutions, while method 2 ("INDELible-m2") uses instead the Gillespie approach for the same task. Here we consider both algorithms. Seq-Gen [17] is a particularly popular and efficient simulation tool, but it does not allow simulation under complex models, and in particular it does not allow simulation of indels. Finally, we consider the software pyvolve [36], which offers an extremely broad choice of models of sequence evolution, and, like phastSim, is implemented in Python.

**SARS-CoV-2 datasets.**   As a ground for comparison between existing sequence evolution simulators we consider different scenarios typical for genomic epidemiology. First, we consider the simulation of a scenario similar to SARS-CoV-2 evolution. We simulate trees with a custom script (an adapted version of the simulator NGESH [39]) and under a Yule process with birth rate equal to genome length (29,903), so to have in the order of one mutation per branch. We use these trees as input for different sequence simulation methods, tracking their computational demand. We simulate sequence evolution under an UNREST model [31] with rates inferred from SARS-CoV-2 data [22] where possible (for phastSim, pyvolve [36] and INDELible [18]) and a GTR model [6] otherwise (for Seq-Gen [17]).

While in the basic simulation setting we consider a simple model of nucleotide evolution without substitution rate variation across the genome, without selection and without indels, we also extend this scenario so to measure the impact of different simulation parameters on the performance of different methods. In the scenario of discrete category rate variation, we consider a nucleotide model with 10 categories of substitution rates across the genome. In the

scenario of continuous rate variation we consider a nucleotide model with continuous variation in rate (each site has a distinct rate sampled from a Gamma distribution). In the the codon model scenario we consider a codon substitution model with substitution rates derived from the nucleotide model. In the codon model with discrete rate variation scenario, we consider in addition 10 discrete rate categories across the genome for the selection parameter $\omega$. In the codon model with continuous rate variation scenario, we consider a codon model where each codon is allowed its own value of $\omega$ (only allowed in phastSim). Finally, for methods that allow indels (INDELible and phastSim) we also consider the scenario of a nucleotide sequence evolving with the same UNREST substitution model described before in addition to uniform insertion and deletion rates of 0.1 and with indel length distribution of Geo(0.5).

### Bacterial datasets

To demonstrate a scenario in which we are interested in simulating bacterial genome evolution within one outbreak, we use the *E. Coli* reference genome (https://www.ncbi.nlm.nih.gov/nuccore/U00096.3 [40], 4,641,652 nucleotides) as our root genome sequence. Here we only consider the scenario of a nucleotide model without rate variation. We again assume a scenario typical for genomic epidemiology, that the birth rate of the simulated tree is equal to the genome length. The number of mutations simulated is therefore comparable to the number of branches in the tree.

## Results

### SARS-CoV-2 scenario

Due to the vast size of current SARS-CoV-2 genome datasets, simulation of SARS-CoV-2 genomes is expected to be an application where simulator efficiency will be particularly important. While phastSim and pyvolve are both Python implementations, therefore sharing similar benefits (high compatibility with other packages and ease of extensions) and draw-backs (reduced efficiency compared to some other languages), we see that the two approaches have dramatically different run time demands (Fig 3): simulating 50 sequences under pyvolve requires on average more time than simulating 500,000 in phastSim. We can also see that INDELible-m2 is marginally faster than INDELible-m1 in this scenario, due to the low number of mutations per branch. However, while phastSim and INDELible-m2 are both similar Gillespie approaches, simulating 5,000 sequences with INDELible-m2 requires slightly more time than simulating 500,000 sequences in phastSim (Fig 3), despite the fact that INDELible is coded in C++. Seq-Gen appears to be very efficient, but it's still more than one order of magnitude slower than phastSim on large phylogenetic trees in this scenario. Also note that, for large trees considered here, we can reduce computational demand in phastSim by more than 5-fold by not producing a FASTA output alignment; this way we can also save very significant amounts of memory demand. Regarding small trees ($< 10^4$ tips) most of the demand in phastSim is associated with initializing the simulations (loading packages and initializing the genome search tree structure); these initialization costs do not depend on tree size, and instead depend on genome size, and they are why phastSim is relatively less efficient on small trees. If simulation on small trees are indeed of interest, these initialization costs could be reduced by re-using the same genome search tree structure over multiple replicates, or, in the case of simple evolutionary models, by using our simple, non-hierarchical simulation approach. Differences in memory demands between methods are similar to time demands, with the benefits of phastSim being even more evident (S1 Text).

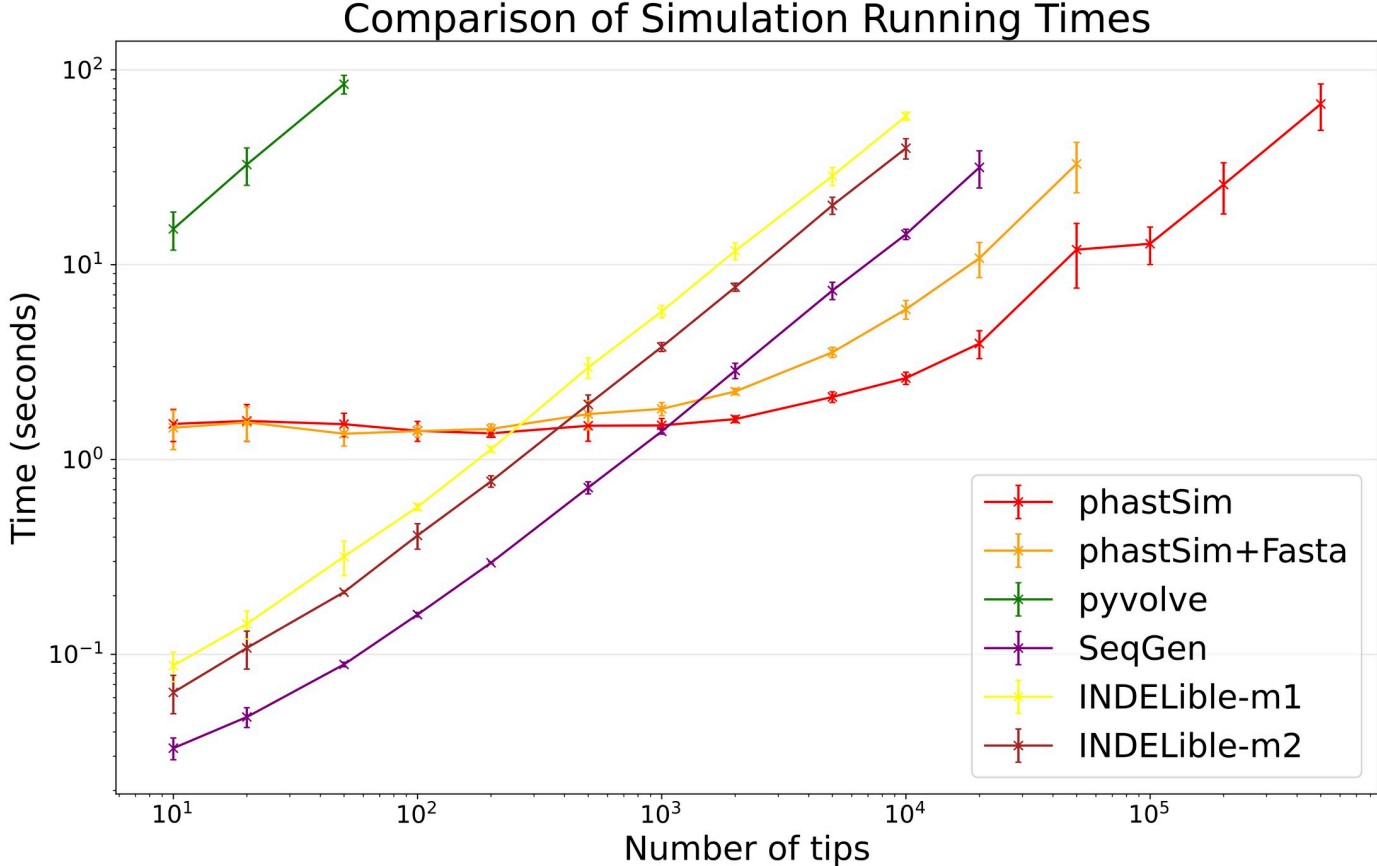

**Fig 3. Comparison of running times of different simulators in a scenario similar to SARS-CoV-2 data.** On the Y axis we show the number of seconds it takes to perform simulations using different software. On the X axis is the number of tips simulated. Each point represents ten replicates. We do not run the most demanding simulators when each replicate would take substantially more than 1 minute to run. In red is the time to run phastSim with a concise output, and in orange is the time for phastSim with additionally generating a FASTA format output. In green is the demand of pyvolve, and in purple of Seq-Gen. In yellow and brown are respectively the time for running INDELible with method 1 (matrix exponentiation) and method 2 (Gillespie approach).

## Bacterial scenario

To showcase the impact of simulated genome size on the performance of different methods, we consider the simulation of the evolution of an entire bacterial genome. As genome size increases, time and memory demand of traditional simulators is expected to grow linearly. Indeed, we now see that Seq-Gen takes considerably more time to simulate the same number of genomes than in the SARS-CoV-2 scenario (Fig 4). phastSim also has an increased computational demand, but only in terms of the initial step of generating an initial genome search tree. This initial cost is linear with respect to genome length, but does not increase with the number of samples or with the number of mutations simulated. In total, in this scenario phastSim can simulate sequence evolution along trees with more than 1000 times more samples than Seq-Gen. A further reduction in computational demand, in particular in terms of the initial cost of generating a genome search tree, can be obtained by using the simple non-hierarchical algorithm (Fig 4), which however comes at the cost of narrowing the choice of evolutionary models to less complex ones. Similar results are observed with respect to memory demand (S1 Text).

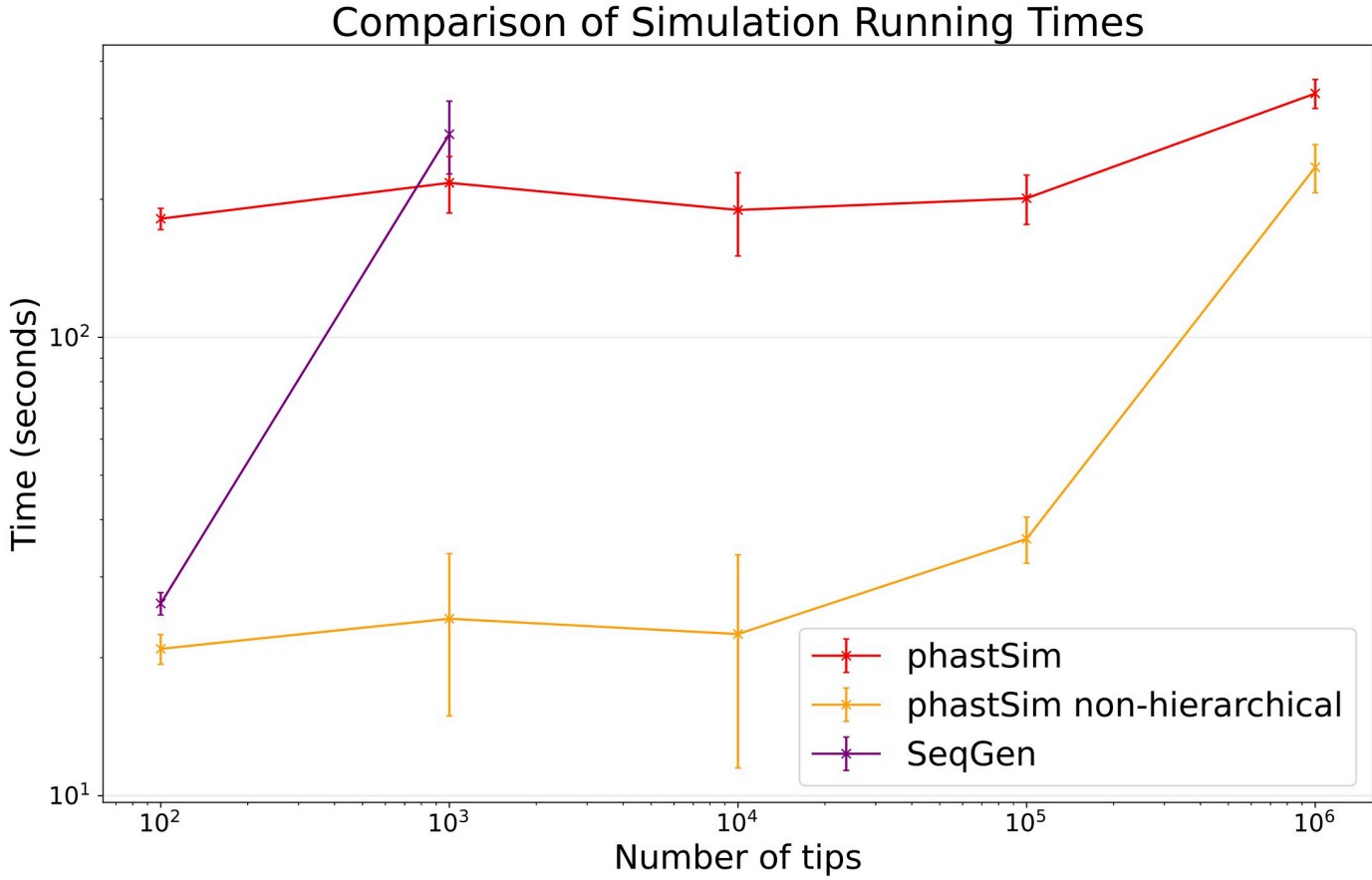

**Fig 4. Comparison of running times of different simulators in a scenario similar to *E. Coli* outbreak data.** On the Y axis we show the number of seconds it takes to perform simulations using different software. On the X axis is the number of tips simulated. Each point represents ten replicates. We do not run Seq-Gen for more than 1000 tips due to high computational demand. In red is the time to run phastSim, and in orange is the time for phastSim with the simple, non-hierarchical approach. In purple is the time demand of Seq-Gen.

## Evolutionary and indel models

One of the advantages of the approach we present here is that simulating evolution under increasingly complex models comes at almost no additional time or memory cost (Fig 5 and S1 Text). It can be seen, for example, that INDELible-m1 and Seq-Gen incur a significantly higher time cost when using a continuous variation in mutation rate. Running INDELible-m2 with a codon model appears to come with no additional computational demand, similarly to phastSim (Fig 5). For these comparisons we have considered the SARS-CoV-2 simulation scenario.

Our algorithm also allows efficient simulation of insertion and deletion events (indels). Among the other simulators considered here, only INDELible can simulate indels. In the SARS-CoV-2 scenario, phastSim can simulate substitutions and indels under about 10 times larger phylogenies than INDELible for the same computational run time (Fig 6), and under about 200 times larger phylogenies for the same memory demand (S1 Text).

## The impact of branch lengths

Probably the main limiting factor in the applicability of the approach presented here are tree branch lengths. Since the demand of our approach is affected linearly by the number of

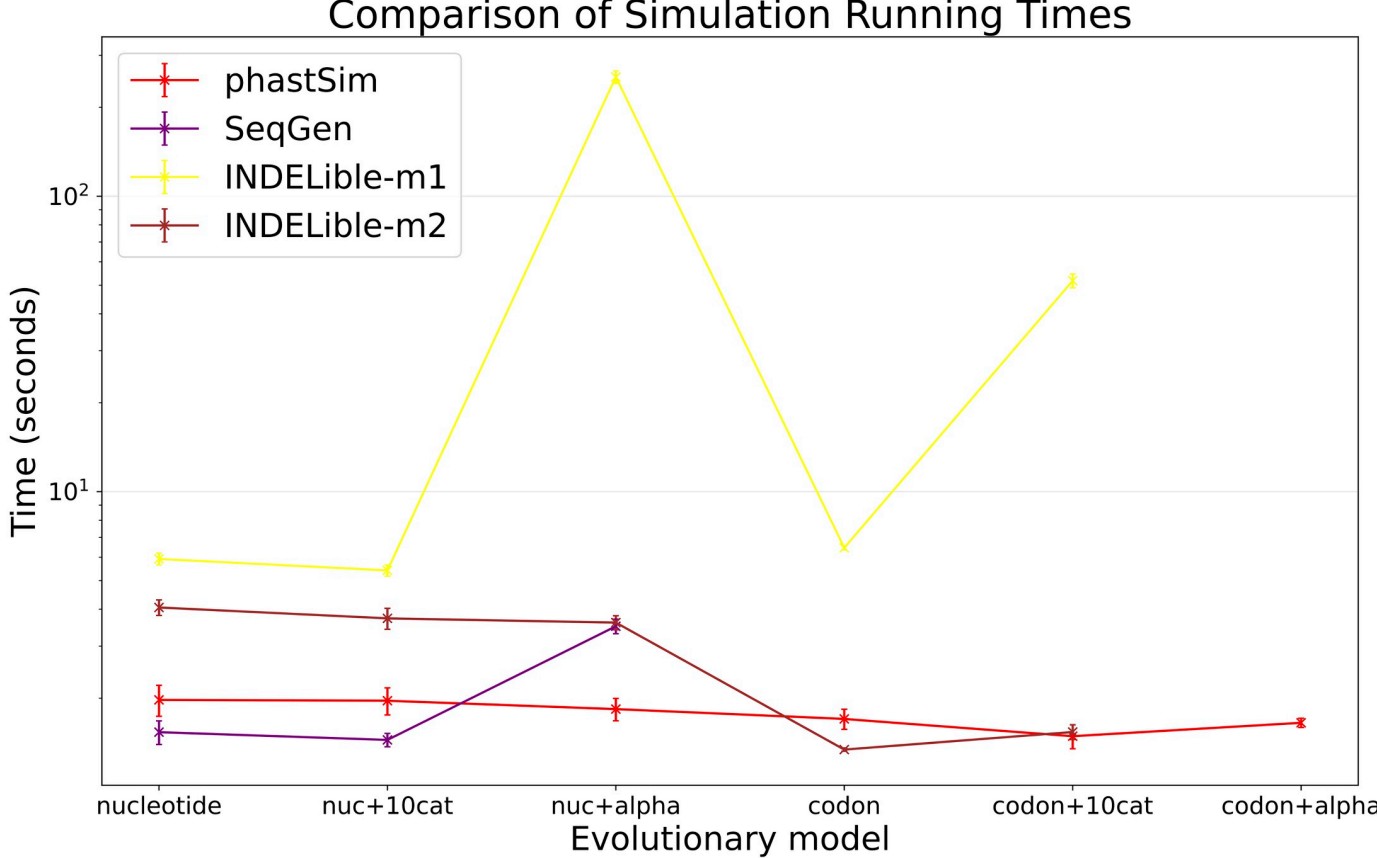

**Fig 5. Comparison of running times of different simulators in a SARS-CoV-2 scenario using different evolutionary models.** On the Y axis we show the number of seconds it takes to perform simulations using different software. On the X axis is the model used for simulations: "nucleotide" is a nucleotide substitution model without variation; "nuc+10cat" is a nucleotide model with 10 rate categories; "nuc+alpha" is a nucleotide model with continuous variation in rate (each site has a distinct rate sampled from a Gamma distribution); "codon" represents a codon substitution model; "codon+10cat" represents a codon substitution model with 10 categories for $\omega$; "codon+alpha" is a codon model with continuous rate variation in mutation rate and in $\omega$ (only allowed in phastSim). Each value represents ten replicates. Seq-Gen does not allow codon models. Colors are as in Fig 3. Here we used alignments of 1000 tips.

mutation events, and as we scale up the length of the tree we need to simulate more mutation events, then the length of the phylogenetic branches will significantly affect the performance of our approach. We can see that, in the SARS-CoV-2 scenario, the impact is not strictly linear (Fig 7). This is because there are additional factors which contribute to phastSim demand in addition to the number of mutation events. For example, one also has to consider the time to initialize the genome search tree, which is linear in genome size, as well as the time to read, initialize, and traverse the input phylogenetic tree, which are linear in the number of tips. Predictably, the computational demands of Seq-Gen and INDELible-m1 seem not affected by the length of the branches. It is instead surprising to see that the computational demand of INDELible-m2 seems also not affected by the branch lengths, despite it using a Gillespie approach; the reason is that probably other factors, independent of the number of mutations, cause the bulk of the demand in this scenario.

## Discussion

We have introduced a new approach to simulating sequence evolution that is particularly efficient when used on phylogenies with many tips and with short branches. Our software

## Comparison of Simulation Running Times

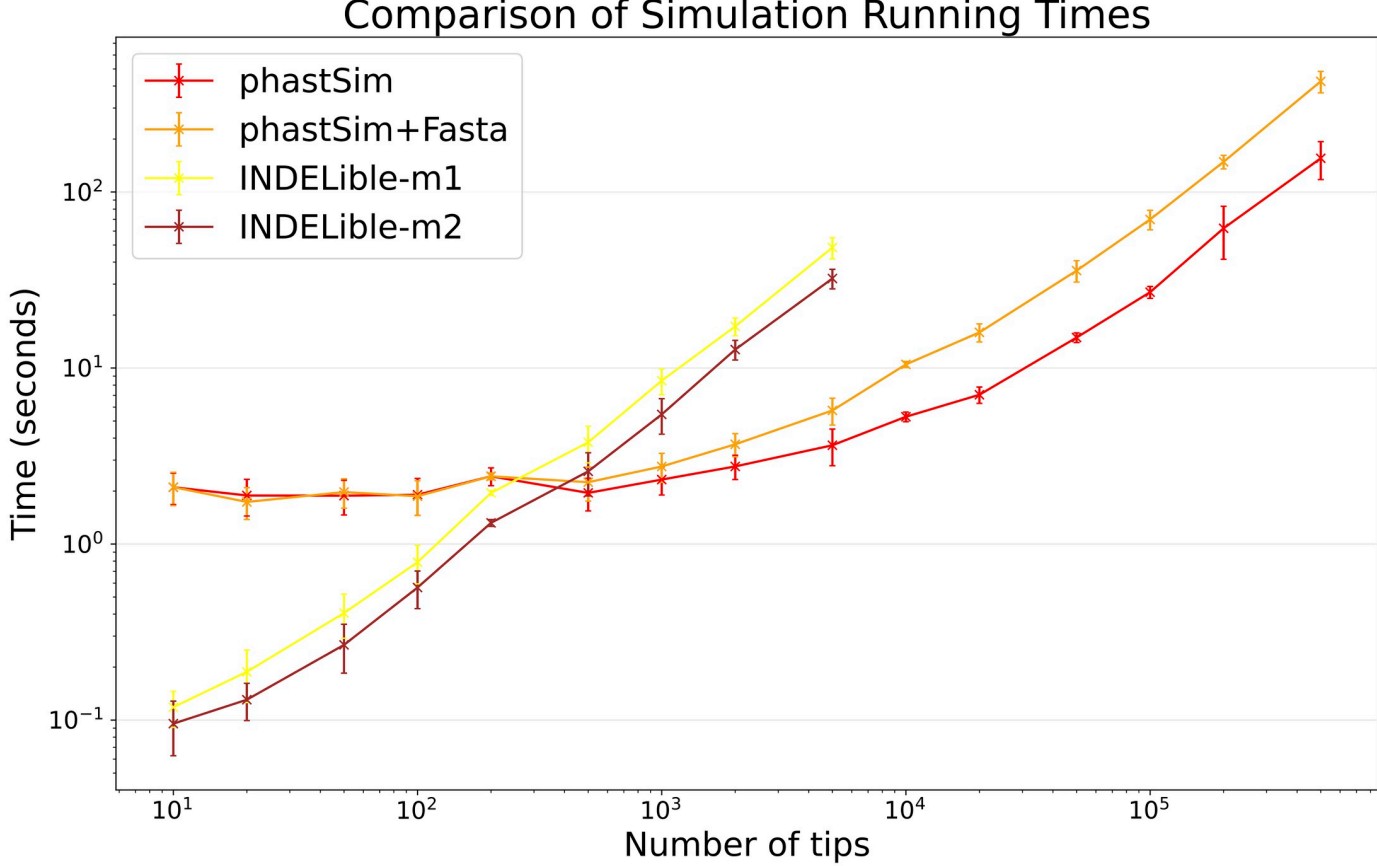

**Fig 6. Comparison of running times of Indelible and phastSim simulators in a SARS-CoV-2 scenario with indels.** In this scenario we compare phastSim against Indelbile-m1 and Indelible-m2 (the only other methods considered here that model indels). Each value represents ten replicates.

phastSim implements this new algorithm and is implemented in Python, allowing it to be easily extended and combined with other Python packages. phastSim relies on the ETE 3 tree phylogenetic structure, and in particular it uses ETE 3 to read input phylogenetic trees. This allows flexibility in the phylogenetic tree input format. Furthermore, thanks to the fact that the efficiency of the algorithm is not affected by the complexity of the substitution model used, we allow a broad choice of evolutionary models, such as codon models with position-specific mutation rates and selective pressures. We also implement a new model of hypermutability to more realistically describe the mutational process in SARS-CoV-2. Also, we can efficiently simulate indel events, which are rarely modeled by other simulation packages.

We show that, compared with other simulators, phastSim is more efficient in the scenarios common to genomic epidemiology, that is, when simulating many closely related bacterial or viral genomes. Its particular efficiency with bacterial genomes means that it ideally matches the needs of software that simulate bacterial ancestral recombination graphs (e.g. [9, 41]). phastSim can also be easily run using the output of phylogenetic simulator, most relevantly VGsim [42] which allows fast simulations of very large and short phylogenies typical of SARS-CoV-2 and other genomic epidemiological scenarios, and which also allows the simulation of the effects of selection on the phylogenetic tree shape. phastSim is implemented as a Python package, which allows for easy integration into other Python pipelines.

## Comparison of Simulation Running Times

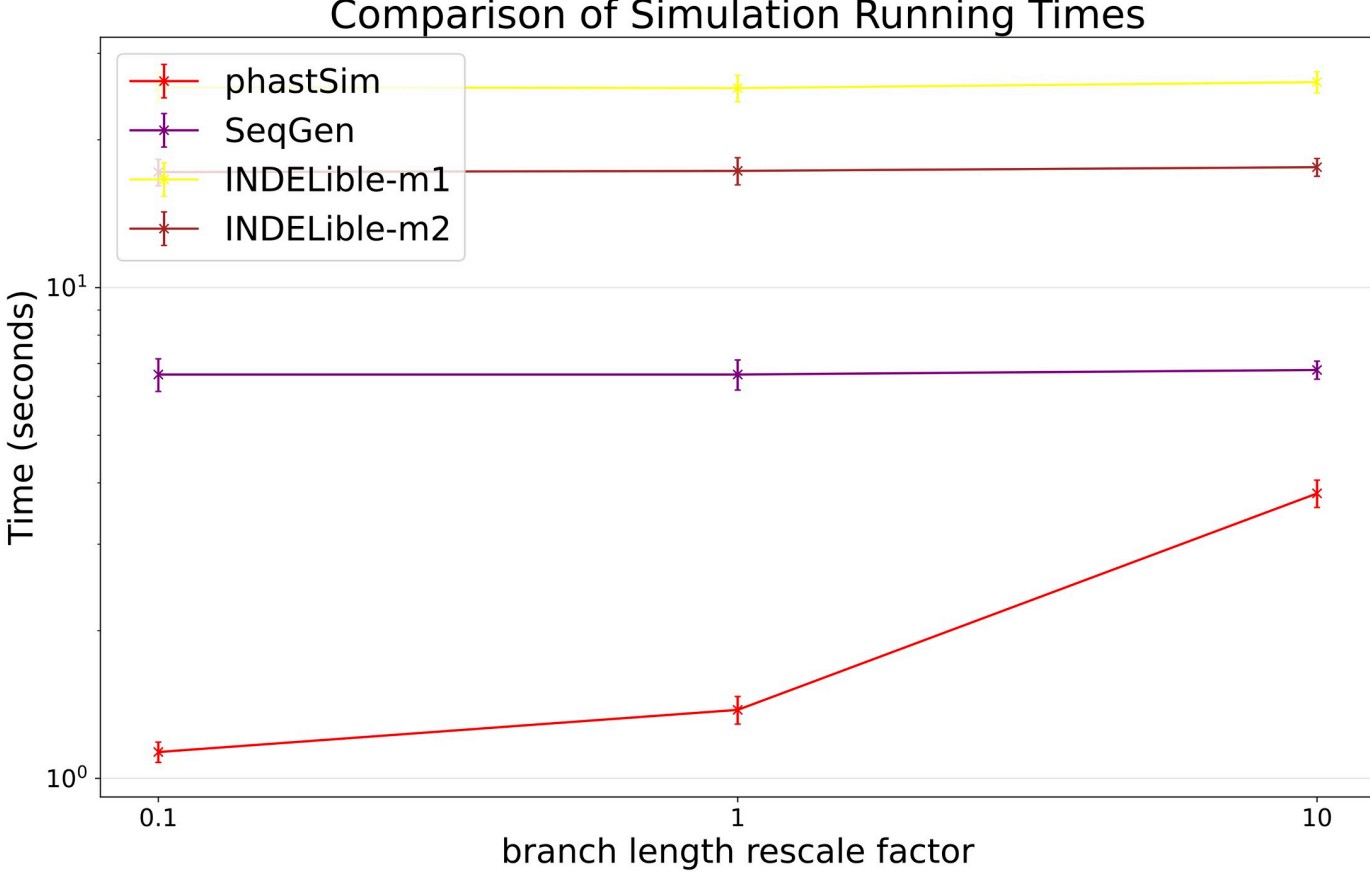

**Fig 7. Comparison of running times of different simulators in a SARS-CoV-2 scenario after rescaling the tree branch lengths by different factors.** On the Y axis we show the number of seconds it takes to perform simulations using different software. On the X axis is the rescaling factor we use to make the phylogenetic tree branch lengths longer or shorter. Colors are as in Fig 3. Here we used alignments of 5000 tips.

In the future, it would be possible, and of interest, to expand the features of phastSim, in particular allowing a broader spectrum of models, for example allowing column-specific amino acid fitness profiles; also, it could be possible to implement the described algorithm in more efficient programming languages.

In conclusion, we have presented a novel algorithm, and corresponding software implementation phastSim, to efficiently simulate sequence evolution along large trees of closely related sequences. This new approach considerably outperforms other methods in the scenarios of genomic epidemiology, for example when simulating SARS-CoV-2 genome sequence datasets. This approach also allows for more realistic models of sequence evolution, allowing more efficient and accurate sequence data simulation and inference.

## Supporting information

**S1 Text. Supplementary text.** Supplementary file containing results regarding memory demand of different methods and containing results from testing of phastSim features. (PDF)

## Acknowledgments

We are very thankful to Vladimir Shchur for the valuable suggestions on our work.

## Author Contributions

**Conceptualization:** Nicola De Maio.

**Data curation:** Nicola De Maio.

**Formal analysis:** Nicola De Maio, William Boulton.

**Funding acquisition:** Nick Goldman.

**Investigation:** Nicola De Maio, William Boulton.

**Methodology:** Nicola De Maio, William Boulton.

**Project administration:** Nicola De Maio, Nick Goldman.

**Software:** Nicola De Maio, William Boulton, Lukas Weilguny, Conor R. Walker.

**Supervision:** Nicola De Maio, Nick Goldman.

**Validation:** Nicola De Maio, William Boulton.

**Visualization:** Nicola De Maio, William Boulton.

**Writing – original draft:** Nicola De Maio, William Boulton.

**Writing – review & editing:** Nicola De Maio, William Boulton, Lukas Weilguny, Conor R. Walker, Yatish Turakhia, Russell Corbett-Detig, Nick Goldman.

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
