## [Decision Letter · Decision Letter 0]

10 Nov 2021

Dear Dr De Maio,

Thank you very much for submitting your manuscript "phastSim: efficient simulation of sequence evolution for pandemic-scale datasets" for consideration at PLOS Computational Biology.

As with all papers reviewed by the journal, your manuscript was reviewed by members of the editorial board and by several independent reviewers. In light of the reviews (below this email), we would like to invite the resubmission of a significantly-revised version that takes into account the reviewers' comments.

I agree with the consensus among the three reviewers. This approach is novel and potentially quite useful. That said, the reviewers identified several areas that require attention and improvement. In particular, Reviewers #1 and #3 raise an important point about the structure and focus of the manuscript.

We cannot make any decision about publication until we have seen the revised manuscript and your response to the reviewers' comments. Your revised manuscript is also likely to be sent to reviewers for further evaluation.

Sincerely,

Joel O. Wertheim

Associate Editor

PLOS Computational Biology

Ville Mustonen

Deputy Editor

PLOS Computational Biology

I agree with the consensus among the three reviewers. This approach is novel and potentially quite useful. That said, the reviewers identified several areas that require attention and improvement. In particular, Reviewers #1 and #3 raise an important point about the structure and focus of the manuscript.

Reviewer's Responses to Questions

**Comments to the Authors:**

Reviewer #1: In this manuscript, the authors present a novel tool for simulating the evolution of an ancestral sequence along a given phylogeny. The authors have made the tool available as an open source package available on GitHub with user-friendly installation possible via PyPI (using the pip package manager).

As promised in the manuscript and GitHub repo, I was able to easily install the tool, along with all of its dependencies, on my laptop via a simple "pip install phastSim" command (using an Ubuntu 18.04 environment in the Windows Subsystem for Linux). I was then able to run the tool using the example dataset provided in the GitHub repo, which finished running in just ~15 seconds on my laptop, which is quite impressive given the sample dataset size. I only briefly skimmed through the code base on GitHub, but at a glance, it's quite clean and organized. Regarding the algorithms presented, the authors present clever techniques for efficiently simulating sequence evolution along a tree, and importantly, their approach supports the simulation of insertions and deletions, something not supported by Pyvolve nor (if I recall correctly) Seq-Gen.

Regarding the manuscript itself, I believe the paper is overall well-written: as expected from a paper of this nature, the authors (1) introduce the bioinformatics problem at hand, (2) discuss prior work in the space, (3) introduce their novel approach, (4) present their approach's algorithms and tool implementation, (5) describe a simulation experiment to benchmark their tool against existing methods, (6) present the results of the benchmarking experiment, and (7) discuss the results. However, I believe the paper requires some significant revision:

- The technical details of the simulation experiment are currently presented in the "Results" section of the manuscript. From my perspective, it would make more sense to move the technical details about the methods behind the simulation experiment to the "Materials and methods" section of the manuscript. I believe only the results of the simulation experiment (i.e., the actual benchmarking measurements) should be presented in the "Results" section

- The paper only shows runtime measurements for the various tools, but because of the large number of simulations that need to be executed, ideally with many replicates in parallel, in large-scale simulation experiments such as those used to study COVID-19 (e.g. Pekar et al., Science 2021), and because the simulation of sequence evolution can be quite memory-intensive (as mentioned by the authors), the benchmarking results should include plots depicting peak memory usage of the various tools as well. I apologize in advance for asking for this, as I'm sure it'll require quite of work to be redone, but in addition to runtime measurements, peak memory measurements are critical to properly compare these tools

- Figure 1 should be cleaned up to look a bit more professional / production-quality. For example, the child branches coming out of the internal nodes of the tree are quite inconsistent in terms of spacing, and rather than using "->" to denote a right arrow, it would be better to actually use a right arrow (→), etc.

- Figure 2 should be cleaned up substantially: the image looks quite distorted (namely the small red-and-blue trees), perhaps because of resizing vertically?

- Figures 4, 5, and 7 should be redone to look consistent with Figures 3 and 6 (especially the legends and tick labels). Further, these 3 figures have far too much vertical space: the y-max should be much smaller (e.g. 150 seconds for Figure 4, 3 seconds for Figure 5, and 45 seconds for Figure 7)

- Why are Figures 4, 5, and 7 using different-sized trees for each tool in these experiments? These should be replaced with the exact same trees for each tool, just as was done in Figures 3 and 6. If the decision to use different-sized trees was for the sake of presentation due to huge variation in runtime across the tools, the authors can use a log-scale for the vertical axis. Even in the figures' current form, the boxes are quite squished, and log-scale may help better depict them

- Figure 5's minimum vertical axis value (y-min) should be 0, not -1, as these are runtimes (if the vertical axis is changed to log-scale as I recommended in a prior bullet, the y-min would need to be a positive number rather than 0)

- I was not able to find the datasets used in the simulation experiments. I understand that GISAID has tight restrictions on releasing actual sequences, but the phylogenies used in the simulation experiments, along with the raw benchmarking measurements should be made publicly available (e.g. in a separate GitHub repo, on Data Dryad, on figshare, etc.). If the authors are worried about GISAID terms with respect to the phylogeny, the only identifiable component would be the tip labels, so the authors can simply replace the tip labels with arbitrary values (e.g. "0", "1", etc.). I would recommend also including all scripts/commands utilized in conducting the benchmarking experiments so that a reader can simply copy-and-paste the exact commands you used and (more-or-less) reproduce the benchmarking results

Less significant general comments for improvement of presentation:

- The formatting of the pseudocode in the various algorithms is somewhat inconsistent. Of note, the spacing between the equal signs in assignments is inconsistent (sometimes "a=b", sometimes "a= b", sometimes "a =b", and sometimes "a = b"). It would be good to revise to be consistent; I would recommend putting spaces between symbols for clarity

- Assignment operations in pseudocode are typically denoted using a left arrow (←) rather than using an equal sign (=)

- Multiple parts of the paper say "sample ___ from an exponential distribution with parameter ____", which is slightly ambiguous: the exponential distribution has two possible parameterizations (rate, or scale = 1/rate), and while the rate parameterization is the most typical representation (to my knowledge), it would be good to specify, e.g. "sample ____ from an exponential distribution with rate parameter ____"

- All figures appear quite pixelated in the PDF I downloaded. However, this may be an artifact of the submission system, so it may not actually be an issue on the authors' end (but it would be good to double check)

- Figure 3 may be improved by presenting the vertical axis in log-scale to better distinguish between the runtimes of smaller values of "number of tips" (though not necessary, as the trends are quite clear even in the current presentation)

Specific comments about wording/grammar/text throughout the manuscript:

- "Sequence simulators are fundamental tools in bioinformatics, as they allow us to test data processing and inference tools, as well as being part of some inference methods"  The last clause of this sentence is grammatically incorrect and should be revised

- "Here we present a new algorithm and software for ..."  There should be a comma after "Here"

- "Our algorithm is based on the Gillespie approach, and implements an ..."  There should be an "it" before "implements"

- "either for example through Approximate Bayesian Computation [6, 7], see e.g. [8, 9],"  I wonder if this could just be changed to be "Approximate Bayesian Computation [6-9]" (i.e., remove the "e.g." part)? Same comment for the following sentence

- The paragraph starting with "In this simplified “vanilla” scenario..." may benefit from being split into two parts, e.g. with a new paragraph starting at " A pseudocode description of..."

- The end of the paragraph starting with "In this simplified “vanilla” scenario..." presents details about the tool implementation, though those tool-specific descriptions should likely be moved to the portion of the manuscript that describes the tool

- There are some more minor grammar issues throughout, so the paper may benefit from another pass of internal revisions for such things

Overall, I was thoroughly impressed by this work, and I look forward to utilizing phastSim in my own research!

Reviewer #2: The review is uploaded as a separate PDF.

Reviewer #3: Summary. The authors develop phastSim, a software package for simulating the evolution of sequences along a tree. The authors' primary innovation is the development of a data structure that allows efficient computation of sequences on large trees. As someone who is an expert in this field, I have experimented with using a binary-search tree to efficiently identify the location of a mutation during simulation. I also abandoned such an algorithm because of the primary problem diagnosed in this paper: copying the binary-search tree to descendant phylogenetic branches is an expensive operation. The authors solved this problem by developing a multi-layer binary search tree that doesn't have to be copied at every phylogenetic split. Instead nodes maintain different views of the shared data-structure, and descendant branches add nodes to the data-structure to update their views when mutations happen without affecting other views. I found this algorithm an interesting solution to the problem.

Major Comments. The paper's primary result is comparing features and runtime between existing programs. Such comparisons should be a secondary result in my opinion. Instead, simulation papers should focus on demonstrating the accuracy of their simulation software. However in this paper, there is no evidence presented that the simulated data generated by phastSim agrees with the models being simulated. It's not uncommon to find subtle bugs in simulation programs that introduce bias into simulations. This is why it is important for simulation papers to demonstrate their accuracy before they compare their performance to other programs. Accuracy can be demonstrated several ways, including using summary statistics, statistical tests, or parameter estimation to show that the simulated output matches what one would expect from the model. Doing all three for several different models support by phastSim would make a strong case that the software is accurate.

It appreciate that the code is open source and freely licensed.

Minor Comments. Several of the algorithms presented as figures in the manuscript were adequately explained in the text. I think the paper would be improved by removing some of these algorithms from the paper. For example, Algorithms 2 and 6.

**Have the authors made all data and (if applicable) computational code underlying the findings in their manuscript fully available?**

Reviewer #1: Yes

Reviewer #2: Yes

Reviewer #3: Yes

PLOS authors have the option to publish the peer review history of their article (what does this mean?). If published, this will include your full peer review and any attached files.

Reviewer #1: **Yes: **Niema Moshiri

Reviewer #2: No

Reviewer #3: No
---

## [Decision Letter · Decision Letter 1]

14 Mar 2022

Dear Dr De Maio,

Thank you for the thorough revision. Please have a look of the few remaining, mostly stylistic, suggestions by the reviewers and try to accommodate them.

Thank you very much for submitting your manuscript "phastSim: efficient simulation of sequence evolution for pandemic-scale datasets" for consideration at PLOS Computational Biology. As with all papers reviewed by the journal, your manuscript was reviewed by members of the editorial board and by several independent reviewers. The reviewers appreciated the attention to an important topic. Based on the reviews, we are likely to accept this manuscript for publication, providing that you modify the manuscript according to the review recommendations.

Sincerely,

Ville Mustonen

Deputy Editor

PLOS Computational Biology

Ville Mustonen

Deputy Editor

PLOS Computational Biology

[LINK]

Reviewer's Responses to Questions

**Comments to the Authors:**

Reviewer #1: We thank the authors for their significant revisions. The paper looks excellent, and all of our key concerns have been addressed. We have the following (extremely minor) comments regarding the revised version of the manuscript:

- In Figures 3 and 6, it's unclear to me why "tree generation" is included in the runtime comparisons, as it's not really relevant to the task at hand (sequence simulation). I would recommend removing it such that the comparison is only between sequence simulation methods

- In Figure 3, in the blue curve (tree generation), why is there such a large variance at the 5th point from the left? I imagine at least 1 measurement may have gotten skewed by background processes on the benchmarking machine or something; I would recommend trying to rerun that point while the machine is not being used. Note that this comment is moot if the "tree generation" curves are removed from the figures as per my previous comment

- In the Algorithm 6 pseudocode, at the top of the "else" statement, rather than using the syntax "int(l/2)" (which is likely the Python code that was used to typecast the result of a floating point division to int), I would recommend using the mathematical notation for "floor", e.g. ⌊l/2⌋. In general, it may be good for the authors to take a pass through the algorithms to ensure that they are using standard mathematical pseudocode syntax rather than Python-like syntax where applicable

Reviewer #2: The authors have addressed all my comments and I am happy to recommend acceptance at this stage.

Reviewer #3: The authors have fully addressed my comments from the previous version.

I looked at the supplemental figures showing the accuracy of the simulations. I feel that these figures were quickly put together, and the supplement would benefit from some more time spent on them. This includes (1) increasing font sizes following journal guidelines and (2) consistently marking location of no-error on all histograms. Also the histograms seem a bit blocky to me. The authors should explore other visualizations, including jitter plots, qqplots, and empirical CDFs. The authors should also make note of when two lines or plots are on top of one another, that helps readers know that data hasn't been left out.

**Have the authors made all data and (if applicable) computational code underlying the findings in their manuscript fully available?**

Reviewer #1: Yes

Reviewer #2: Yes

Reviewer #3: Yes

PLOS authors have the option to publish the peer review history of their article (what does this mean?). If published, this will include your full peer review and any attached files.

Reviewer #1: **Yes: **Niema Moshiri

Reviewer #2: No

Reviewer #3: No

Figure Files:

Data Requirements:

Reproducibility:

References:

---

## [Editor Report · Decision Letter 2]

25 Mar 2022

Dear Dr De Maio,

We are pleased to inform you that your manuscript 'phastSim: efficient simulation of sequence evolution for pandemic-scale datasets' has been provisionally accepted for publication in PLOS Computational Biology.

Best regards,

Ville Mustonen

Deputy Editor

PLOS Computational Biology

Ville Mustonen

Deputy Editor

PLOS Computational Biology

---

## [Editor Report · Acceptance letter]

26 Apr 2022

PCOMPBIOL-D-21-01738R2 

phastSim: efficient simulation of sequence evolution for pandemic-scale datasets

Dear Dr De Maio,

I am pleased to inform you that your manuscript has been formally accepted for publication in PLOS Computational Biology. Your manuscript is now with our production department and you will be notified of the publication date in due course.

With kind regards,

Anita Estes
